# Longitudinal high-dimensional analysis identifies immune features associating with response to anti-PD-1 immunotherapy

Elaine Lai-Han Leung ®[1,10] ✉, Run-Ze Li[2,3,10], Xing-Xing Fan[4,10], Lily Yan Wang[5], Yan Wang[6], Zebo Jiang[4], Jumin Huang[4], Hu-Dan Pan[2,3], Yue Fan[5], Hongmei Xu[5], Feng Wang[5], Haopeng Rui[5], Piu Wong[7], Hermi Sumatoh ®[8], Michael Fehlings ®[8], Alessandra Nardin ®[8], Paul Gavine[5], Longen Zhou[5], Yabing Cao ®[9] ✉ & Liang Liu ®[2,3] ✉

Response to immunotherapy widely varies among cancer patients and identification of parameters associating with favourable outcome is of great interest. Here we show longitudinal monitoring of peripheral blood samples of non-small cell lung cancer (NSCLC) patients undergoing anti-PD1 therapy by high-dimensional cytometry by time of flight (CyTOF) and Meso Scale Discovery (MSD) multi-cytokines measurements. We find that higher proportions of circulating CD8⁺ and of CD8⁺CD101ʰⁱTIM3⁺ (CCT T) subsets significantly correlate with poor clinical response to immune therapy. Consistently, CD8⁺ T cells and CCT T cell frequencies remain low in most responders during the entire multi-cycle treatment regimen; and higher killer cell lectin-like receptor subfamily G, member 1 (KLRG1) expression in CCT T cells at baseline associates with prolonged progression free survival. Upon in vitro stimulation, CCT T cells of responders produce significantly higher levels of cytokines, including IL-1β, IL-2, IL-8, IL-22 and MCP-1, than of non-responders. Overall, our results provide insights into the longitudinal immunological landscape underpinning favourable response to immune checkpoint blockade therapy in lung cancer patients.

Immune checkpoint blockade (ICB) therapy is known to produce durable clinical responses in a select group of non-small cell lung cancer (NSCLC) patients; therefore, early identification of progression is critical for patient selection. To achieve better effectiveness with ICB therapy, parameters indicating favorable response are critically needed. Several predictors of the response to programmed death-1 (PD-1) blockade have been reported, such as the presence of tumor-infiltrating T cells, high programmed death-ligand 1 (PD-L1) expression in biopsies, microsatellite instability (MSI), KEAP1 and STK11 mutations test and the tumor mutational burden (TMB)[1–4]. However, these

---

[1]Cancer Center, Faculty of Health Sciences; MOE Frontiers Science Center for Precision Oncology, University of Macau, Macau (SAR), China. [2]State Key Laboratory of Traditional Chinese Medicine Syndrome, The Second Affiliated Hospital of Guangzhou University of Chinese Medicine (Guangdong Provincial Hospital of Chinese Medicine), Guangzhou, Guangdong, China. [3]Guangdong-Hong Kong-Macau Joint Lab on Chinese Medicine and Immune Disease Research, Guangdong, China. [4]Dr. Neher's Biophysics Laboratory for Innovative Drug Discovery/State Key Laboratory of Quality Research in Chinese Medicine/Macau Institute of Applied Research in Medicine and Health, Macau University of Science and Technology, Avenida Wai Long, Macao, Taipa Macau (SAR), China. [5]Janssen Research & Development, Shanghai, China. [6]Merck Sharp & Dohme, Shanghai, China. [7]HiFiBio Therapeutics, Hongkong, China. [8]ImmunoScape, Singapore, Singapore. [9]Kiang Wu Hospital, Macau (SAR), China. [10]These authors contributed equally: Elaine Lai-Han Leung, Run-Ze Li, Xing-Xing Fan. ✉e-mail: lhleung@um.edu.mo; sumscaoyabing@hotmail.com; lliu@gzucm.edu.cn

parameters are hampered by the limited longitudinal observation window, the small number of parameters, and the lack of systematic, unbiased bioinformatic pipelines, which has resulted in a paucity of indicators predicting response state to date. Longitudinal investigation is important to trace markers associated with acquired resistance since the anti-PD1-mediated immune response is dynamic during treatment cycles. However, it is unclear whether noninvasive approaches, such as peripheral blood mononuclear cell (PBMC) profiling, can be used to predict responses to ICB therapy by identifying the relevant responding cell types, which could provide insights for understanding the underlying immunological mechanisms of primary and acquired resistance.

Here, we combine high-dimensional CyTOF and multiplex cytokine approaches to longitudinally analyze 25 patients diagnosed with NSCLC over at least 30 months. To better understand the treatment-resistant parameters associated with anti-PD-1 treatment, immunophenotyping, molecular analysis and functional characterization of immune populations are performed. The findings in this study could provide insights to guide precision therapy and elucidate the underlying mechanism of treatment resistance.

## Results

### Enrichment of CD8+ and CCT T subsets predicts the efficacy of anti-PD-1 therapy

To monitor immune cell responses induced by PD-1-targeted therapy, PBMC were isolated and analyzed by CyTOF. Sixteen healthy donors (HD) and 25 patients with NSCLC were enrolled, including 11 responders and 14 non-responders (Fig. S1, Table S1). Patient clinical response was determined on the basis of immune response evaluation criteria in solid tumors (RECIST) criteria (Fig. S2), consistent with published trials in lung cancer[5,6].

To identify the predictive cell subsets, immune cells were subsequently clustered based on 40 leukocyte antigens (Table S2), and the differences between responding and non-responding patients were examined (Figs. S3–S7). Responders had much lower frequencies of CD8+ T cell subtypes, especially CD8+CD101hiTIM3+ (CCT T) cells, which have been associated with exhausted cells[7,8] (Fig. 1a, b). We have tested the relative distributions of CD101 and TIM3 expression in different major immune cell populations. However, there was no statistical significance as shown in Fig. S8. We also tested the differences based on the different histological sub-types of NSCLC. Compared with squamous carcinoma patients, adenocarcinoma patients harbored a higher frequency of exhausted T cells at the baseline, especially in CD8+ CD101hi TIM3+ T cells with $p$-value < 0.05 (Fig. S9).

Longitudinal analysis (Figs. S10 and S11) showed that both CD8+ T cells and CCT T cells remained at low levels in most responders during the 10 treatment cycles (Fig. 1c). We next assessed whether differences in CD8+ and CCT subtype T cells at baseline may be associated with patient clinical outcomes. As shown in Fig. 1d, 38% of the patients with a high frequency (cutoff is median value) of CD8+ T cells had progressive disease (PD), higher than the 28% of the patients with a low frequency of these major immune cells. This phenomenon was more significant for the CCT subtype; 63% of the patients with a high frequency and 23% of those with a low frequency were defined as having PD. Therefore, whether the CCT subtype of CD8+ T cells can serve as a parameter associating with progression-free survival (PFS) in this cohort of NSCLC patients was further studied. As shown in Fig. 1e, patients with a high level of CD8+ T cells exhibited worse PFS (7 months) than those with a low level of CD8+ T cells (13 months). This result indicated that higher frequencies of CD8+ subtypes and CCT T subtype were associated with unfavorable outcomes.

### CD8+ and CCT T cell exhibited specific phenotype in NSCLC patients before and after anti-PD-1 therapy

To explore the treatment-associated changes in phenotype and compare patients with HD, leukocyte antigens related to CD8+ and CCT T cells were comprehensively investigated (Figs. S12–S14). As shown in Fig. 2a, Granzyme B and T-bet, which are functional markers in cytotoxic T cells, were remarkably upregulated in responder CD8+ T cells. A similar trend of Granzyme B and T-bet was observed in CCT T cells (Fig. 2b), in which the CD27 level was significantly decreased. Furthermore, killer cell lectin-like receptor subfamily G, member 1 (KLRG1), which is enhanced in cytotoxic and terminal effector CD8+ T cells[9], was remarkably increased in CCT cells from responders at baseline (Fig. 2c).

Longitudinally (Figs. S15 and S16), the levels of Granzyme B and T-bet were consistently higher in CD8+ and CCT T cells of responders (Fig. 3a, b). In addition, KLRG1 levels in CCT T cells was higher in responders and could effectively discriminate responders from non-responders even at baseline (Fig. 3b). Furthermore, we correlated predictive T cell subtypes with patient tumor clinical characteristics, such as CT scan results for the tumor burden[10]. We applied a practical approach to estimate the tumor burden by using the major measurable tumor lesion on pretreatment PET-CT scan images of NSCLC patients. Indeed, a higher tumor burden was associated with higher frequencies of CD27+CD8+ T cells and CD27+CCT T cells, regardless of whether treatment was given (Fig. 3c). Altogether, these findings indicated that the different phenotypes of CD8+ and CCT T cells between responders and non-responders may be predictive of the clinical efficacy of anti-PD-1 therapy.

### The functional properties of T cells were impaired in non-responders

Next, we correlated cytokine levels with immune subpopulations to determine their functional properties (Fig. 4, Figs. S17–S19). Significantly decreased levels of plasma Eotaxin-3, IL-1β, IL-23, TNF-α, MDC, and MCP-4 were observed in responders versus non-responders (Figs. 4a, S20). Furthermore, we stimulated patient PBMC with a T cell stimulator (anti-CD3/CD28 or PMA) in vitro and collected the supernatant to measure the levels of secreted cytokines. In general, cytokines including IL-1β, IL-2, IL-8, IL-22, and MCP-1 were higher after stimulation in responders compared with non-responders (Fig. 4b and Fig. S21). We further tested CD8+ and CCT T cell cluster-related cytokines in PFS analysis. High levels of IL-22 at baseline was remarkably associated with prolonged PFS (Fig. 4c). The results suggest that the anticancer activity of T cells in non-responders may have been compromised and associated with patient PFS.

## Discussion

To establish reliable criteria for predicting the clinical response to anti-PD-1 therapy, a combination of high-dimensional CyTOF and MSD multiplex cytokine longitudinal analysis was applied to PBMC collected from a cohort of 25 Chinese NSCLC patients who received ICB therapy. To date, this is the longest follow-up time (30 months) in an NSCLC report with high-dimensional investigations. We proposed a noninvasive, feasible approach for tracking the changes in immune subsets, which can help to accurately distinguish responders and non-responders. The repertoire of surface molecules that are preferentially expressed on dysfunctional T cells has been extensively studied, such as TIM3, PD-1, CTLA4, LAG3, and TIGIT[11]. We discovered that the increase of CD8+ T cell frequency is associated with a worse clinical response to anti-PD-1 therapy, which contradicts the conventional understanding that CD8+ T cells are expected to be associated with a favorable prognosis. However, it has been reported that CD8+ T cells transition through the blood and migrate into the tumor site[12]. CD8+ T cell expansion and migration result in a higher number of infiltrating tumor-specific CD8+ T cells, which contribute to clinical responses to

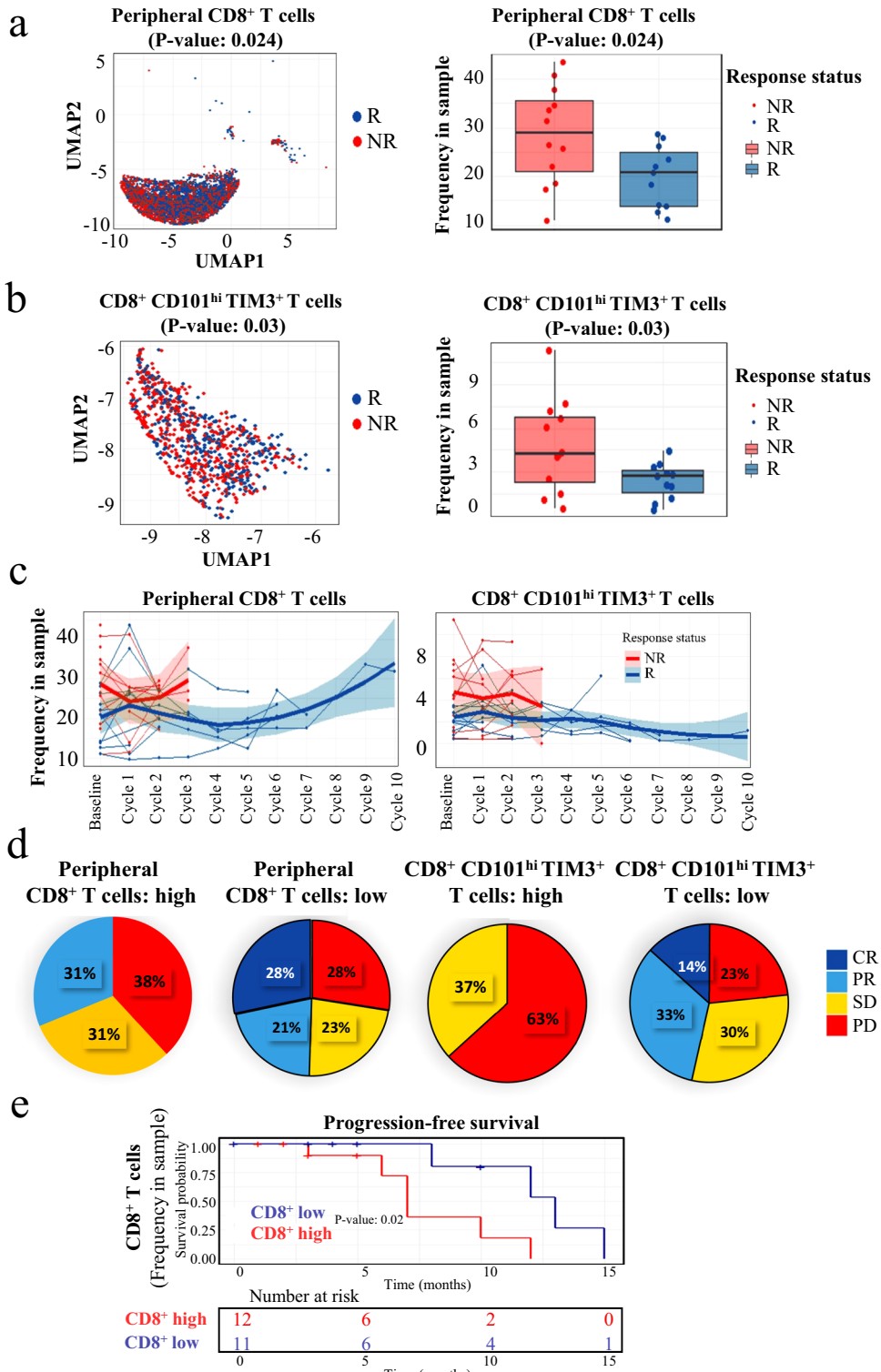

**Fig. 1 | The enrichment of CD8+ and CCT subsets predict the efficacy of anti-PD-1 therapy. a** The frequency of CD8+ T cells from responders' (*n* = 11) PBMC were lower than that of non-responders (*n* = 14) (baseline). This box plots represent the interquartile range (IQR), with the horizontal line indicating the median. The whiskers extend to the farthest data point within a maximum of 1.5 × IQR. **b** Much lower frequencies of CCT subtype were observed in responders' PBMC (*n* = 11) as well (baseline) compared to non-responders' (*n* = 14). This box plots represent the interquartile range (IQR), with the horizontal line indicating the median. The whiskers extend to the farthest data point within a maximum of 1.5 × IQR. **c** By longitudinal analysis, both CD8+ T cells and CCT T cells remained at low levels in most responders during the 10 treatment cycles. Data are presented as mean values

+/− SD. **d** High frequency of CD8+ T cells and CCT T cells was correlated with bad prognosis after the treatment of anti-PD1 therapy. **e** High level of total CD8+ T cell exhibited worse median PFS. The percentages for each subpopulation are based on overall live CD45+ singlet cell counts. *p*-value in PFS analysis was examined by the log-rank test. Other *p*-value was calculated using two-sided t-tests and were corrected for multiple comparisons using the Benjamini-Hochberg adjustment. R responder, NR non-responder, CR complete response, PR partial response, SD stable disease, PD progressive disease. Two baseline samples (J015A and J024A) and 4 samples after treatment (J013B, J029B, and J024 B & C) as they failed the CyTOF QC test were excluded.

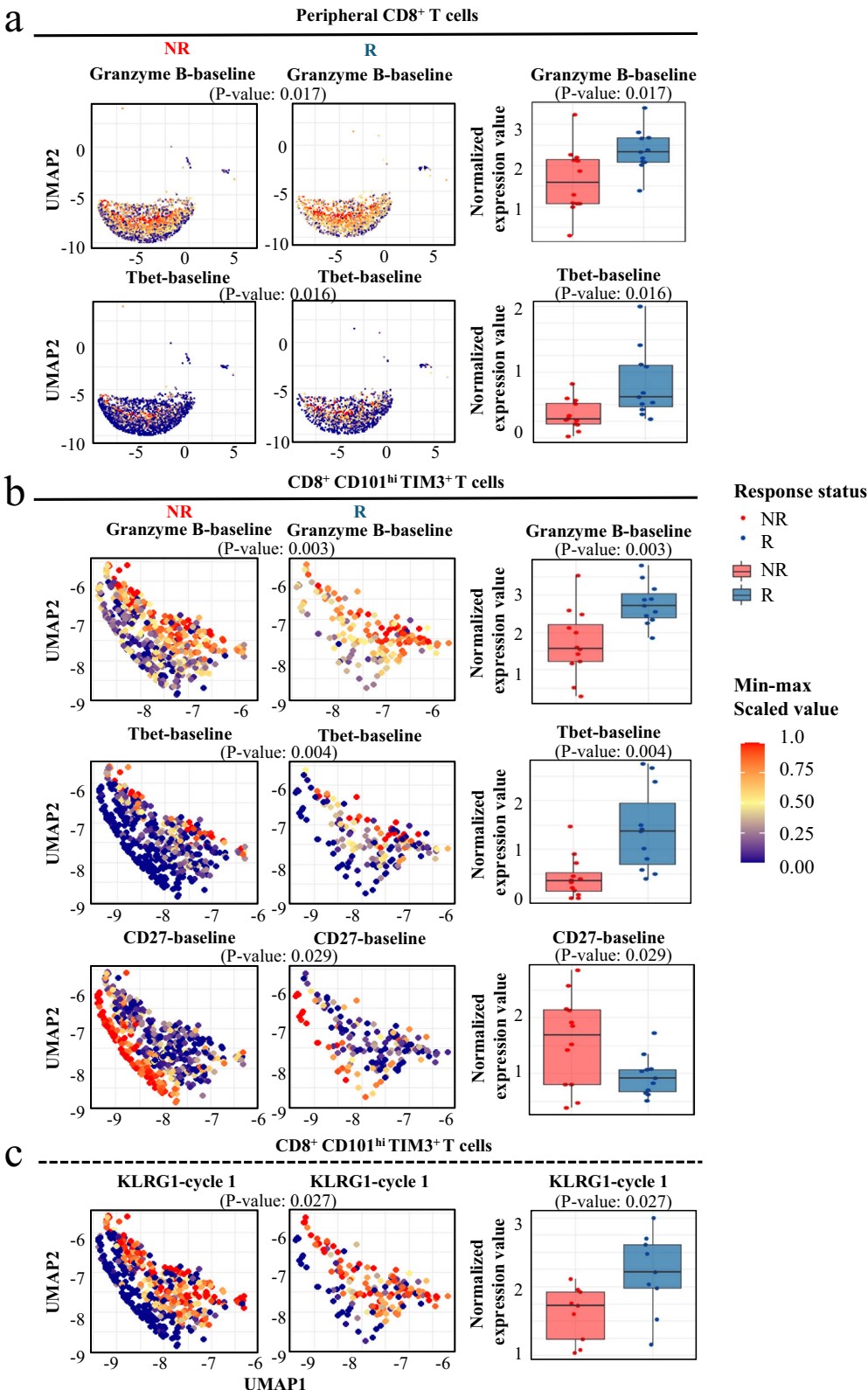

**Fig. 2 | CD8+ and CCT T cells exhibit specific antigens in NSCLC patients before and after anti-PD-1 therapy. a** Granzyme B and Tbet were remarkably upregulated in responders' (*n* = 11) CD8+ T cells compared to non-responders' (*n* = 14). **b** Similar trend of Granzyme B and Tbet were observed in CCT T cells, in which the CD27 level was significantly decreased in responders (*n* = 11) compared to non-responders (*n* = 14). **c** KLRG1 was remarkably increased as well in responders' (*n* = 11) CCT cells compared to non-responders (*n* = 14). The percentages for each subpopulation are based on overall live CD45+ singlet cell counts. All *p*-values were calculated using two-sided t-tests. Two baseline samples (J015A and J024A) and 4 samples after treatment (J013B, J029B, and J024 B & C) as they failed the CyTOF QC test were excluded. R responder, NR non-responder. This box plots represent the inter-quartile range (IQR), with the horizontal line indicating the median. The whiskers extend to the farthest data point within a maximum of 1.5 × IQR.

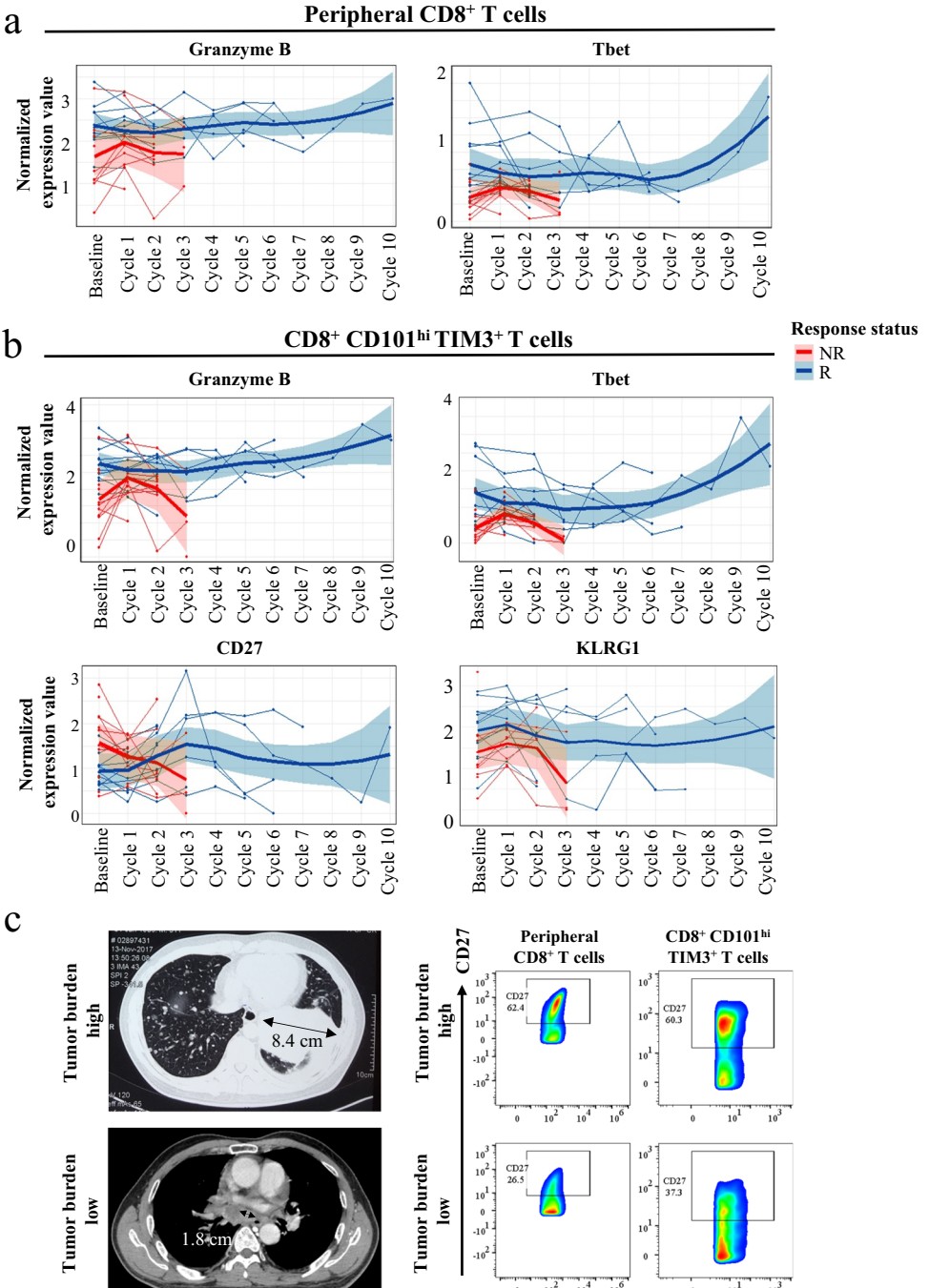

**Fig. 3 | Different phenotypes of CD8+ and CCT T cells between responders and non-responders were predictive of the clinical efficacy of anti-PD-1 therapy.** **a**, **b** Longitudinally the levels of Granzyme B and Tbet were consistently higher in responder CD8+ and CCT T cells. Data in longitudinal analysis are presented as mean values +/− SD. **c** A higher tumor burden was associated with higher frequencies of CD8+CD27+ T cells and CCT T cells. The length of the black arrows corresponds to the tumor size, with the precise value denoted by the black number. The percentages for each subpopulation are based on overall live CD45+ singlet cell counts. R responder, NR non-responder.

PD-1 therapy[13,14]. T cell infiltration can be a positive prognostic indicator in a variety of cancers[15]. Moreover, we identified that the CCT T cell (CD8+CD101hiTIM3+ T cells) subtype was predictive of worse ICB therapeutic efficacy. It has been reported that the irreversible TIM3 and CD101 expression is an hallmark of CD8+ T cells exhaustion during infection[16,17], which is consistent with our results. We observed that the number of dysfunctional CCT T cells was significantly higher in non-responders. Thus, we propose that exhausted T cells would harbor a lower ability to migrate and function.

Another key insight from our research was derived from investigating the connection between cell subsets and cytokine secretion potential. Granzyme B is a serine protease released by CD8+ T cells and natural killer cells and functions as a downstream effector of tumor cytotoxic T cells[18,19]. A recent study found that alterations in the Granzyme B gene (GZMB/H) negatively predicted ICB efficacy in nasopharyngeal cancer patients[20]. Thus, the role of Granzyme B protein as an indicator was investigated in our study. T-bet is an immune cell-specific member of the T-box family of transcription factors and

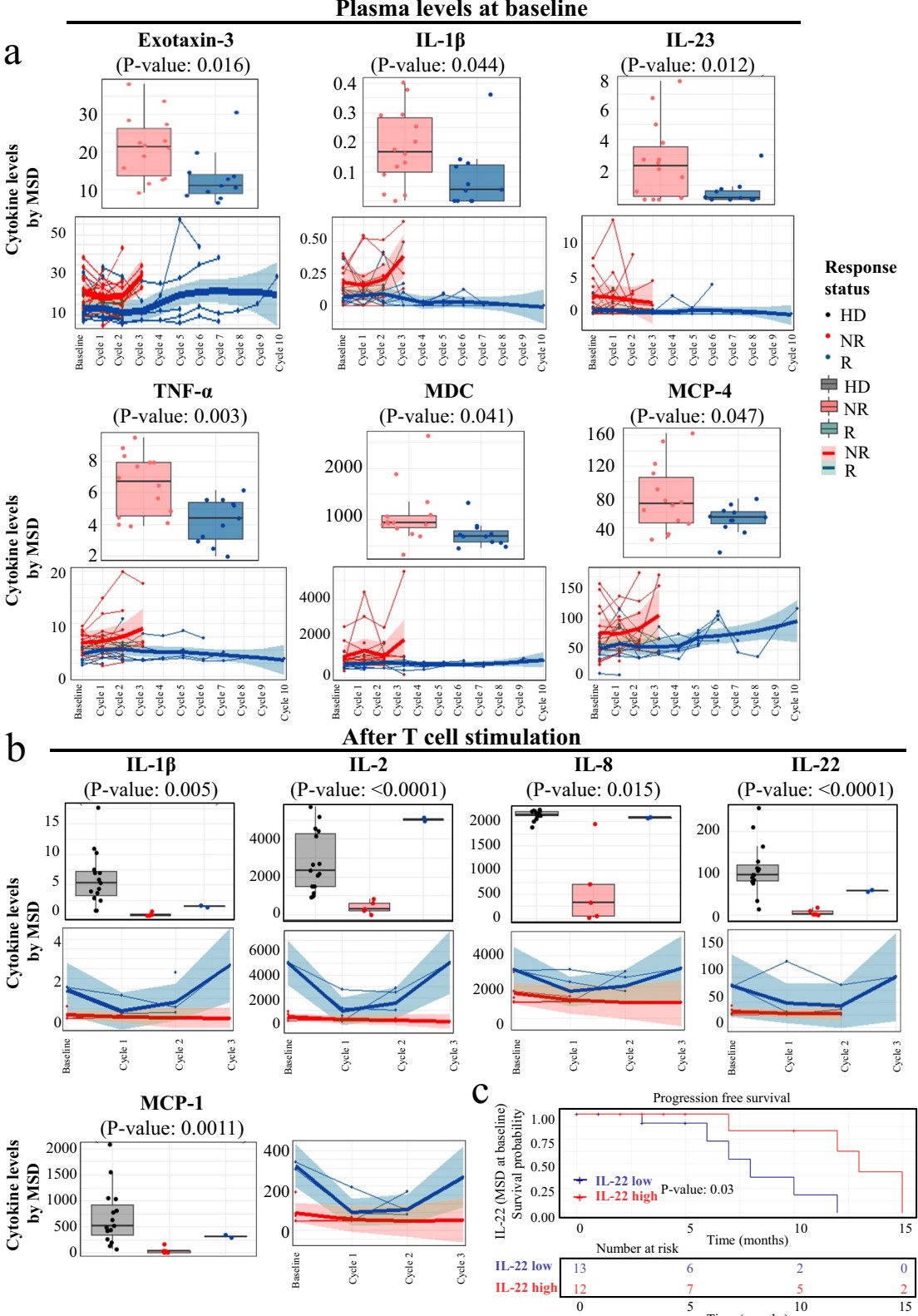

plays a pivotal role in infectious, inflammatory, and autoimmune conditions as the master regulator of effector T-cell activation[21]. It has been reported that patients' overall survival was significantly improved by immunotherapy in patients with high T-bet-expressing T cells[22,23]. We demonstrated here that the levels of Granzyme B and T-bet were higher in CD8+ and CCT T cells of responders (Fig. 3a, b), which suggests indicated the deficiency of these two factors may be linked to

CD8 T cell exhaustion and failure to respond to anti-PD-1 therapy. Most interestingly, the induction of KLRG1 in CCT T cells after one cycle of anti-PD1 treatment was shown to be positively correlated with the efficacy of anti-PD-1 therapy. The function of KLRG1 is well known to be directly involved in T cell activation[9,24]. Consequently, KLRG1+ CCT T cells could represent an activated super killing population with more cytotoxic effector function and a senescent phenotype. To fully

**Fig. 4 | The functional properties of T cells were impaired in non-responders.**
**a** Decreased levels of plasma Eotaxin-3, IL-1β, IL-23, TNF-α, MDC, and MCP-4 were
observed in responders (n = 11) versus non-responders (n = 14). This box plots
represent the interquartile range (IQR), with the horizontal line indicating the
median. The whiskers extend to the farthest data point within a maximum of 1.5 ×
IQR. Data in longitudinal analysis are presented as mean values +/− SD. **b** In sti-
mulated patient PBMC achieved with Dynabeads Human T-Activator CD3/CD28,
secreted cytokines, including IL-1β, IL-2, IL-8, IL-22, and MCP-1, were significantly

increased in responders (n = 11) compared with non-responders (n = 14). This box
plots represent the interquartile range (IQR), with the horizontal line indicating the
median. The whiskers extend to the farthest data point within a maximum of 1.5 ×
IQR. Data in longitudinal analysis are presented as mean values +/− SD. **c** High levels
of IL-22 remarkably contributed to prolonged PFS. p-value in PFS analysis was
examined by the log-rank test. Other p-values were calculated using two-sided t-
tests. HD healthy donors, R responder, NR non-responder. Data are presented as
mean values +/− SD.

understand the immune status in the tumor microenvironment, we
further tested the association of cytokines with anti-PD-1 therapy. IL-1β,
IL-2, IL-22, IL-23, and TNF-α have been demonstrated to be associated
with better prognosis of NSCLC patients.

In conclusion, we have identified a set of immune subtypes that
are enriched in non-responders to ICB therapy. We further elucidated
the connection of immune cells with their correlated cytokines. Unlike
using biopsy, we provide a noninvasive and feasible method for early
and timely monitoring of treatment response. For future clinical uti-
lization, we suggest combining our findings with widely used clinical
parameters to provide more accurate information to predict ICB
treatment efficacy. Moreover, according to our results, this study not
only provided a immunological landscape of response to immune
checkpoint blockade therapy, but also provided a valuable target for
combinational treatment in the future clinical application.

## Methods

### Ethics approval
This study was approved by Kiang Wu Hospital under the approval
number 2018-007.

### Unique Macau lung cancer patients
Patient blood samples and information were collected at Macau Kiang
Wu Hospital, and all patients signed an informed consent form. Total
25 patients were collected. The clinical characteristics of the patients
were shown in Table S1. Among these 25 patients, there were 11
responders and 14 non-responders. The mean ± SD age of responders
was 68.4 ± 11.4 years compared with non-responders' 59.6 ± 11.8 years.
Based on the different histological sub-types of NSCLC, there was a
higher proportion of responders in patients with lung squamous car-
cinoma (75%) compared with the proportion of responders with ade-
nocarcinoma (38.1%). According to the patient demographics and
baseline characteristics, 24 of patients have stage IV, and 1 has stage
IIIB stage. The one patient with stage IIIB received same systemic
treatment as stage IV patients. Based on the patients' PD-L1 IHC level, 4
levels were classified[25]. Then the duration of response time was also
summarized, which showed much longer in responders
(9.4 ± 3.5 months) compared with non-responders (3.6 ± 3.6 months).
For lung cancer patients, the treatment course of pembrolizumab was
dependent on efficacy, but the average cycle was ~3–4 weeks, and the
patients received different treatment courses according to their
treatment efficacy. Clinical pathological data, routine CT scans, and
immunohistochemistry (IHC) data were recorded. Immunological
profiles and patient response correlation analysis were applied. The
percentage of the immune cell subpopulations between the patients
and HD was analyzed. In addition, the correlations of cell subsets with
PD-L1 expression (determined by IHC and routine pathology practice)
and treatment response were analyzed. Ethic code number:
(2018-007).

### Clinical sample collection and PBMC isolation
Whole blood from patients and HD was collected. Blood samples were
stored in EDTA-coated tubes. Human PBMC were isolated using a
density gradient technique (Ficoll-Paque PLUS from GE Healthcare Life
Sciences). Seven milliliters of blood was mixed with 7 ml of a balanced
salt solution, normally PBS. The mixed blood solution was overlayed

on top of 15 ml of Ficoll-Paque PLUS. The mixture was centrifuged at
$300 \times g$ for 15 min at room temperature (RT). During this step, the
granulocytes, platelets, and red blood cells (RBC) pelleted at the bot-
tom of the tube, and the PBMC floated over the Ficoll-Paque PLUS
layer. PBMC were collected from the Ficoll-plasma interface. PBMC
were washed twice with PBS at $300 \times g$ for 5 min. The total number of
PBMC collected from each sample was counted and recorded. Isolated
PBMC were cryopreserved and stored at −80 °C according to a stan-
dard protocol.

### Cytokine assays
The wells of MSD 384-well plates (Meso Scale Discovery) were coated
overnight at 4 °C with 10 μL of coating antibody in carbonate-
bicarbonate coating buffer (15 mM $Na_2CO_3$/35 mM $NaHCO_3$, pH 9.6).
The plates were then washed three times with 35 μL of wash buffer
(0.2% Tween-20 in PBS) per well and blocked with 35 μL of blocking
buffer (2% Probumin/0.2% Tween-20 in PBS) per well for 1 h at RT with
rotational shaking. Plasma and cell culture supernatant samples were
diluted to the desired concentrations in a mixture of aggregation
buffer and at least 50% blocking buffer. After an additional washing
step, 10 μL of sample was transferred to each well of the antibody-
coated MSD plate and incubated with shaking for 1 h at RT. After
aspiration of the samples and four washes with 35 μL of wash buffer
each time, 10 μL of primary detection antibody (shown in the table
below) was added to each well and incubated with shaking for 1 h at RT.
For non-labeled primary detection antibodies, 10 μL of goat anti-
mouse SULFO-TAG-labeled secondary detection antibodies (1:1000 in
blocking buffer; Meso Scale Discovery) was added to each well after an
additional washing step and incubated with shaking for 1 h at RT. After
washing three times with wash buffer, 35 μL of read buffer T with
surfactant was added to each well, and the plate was imaged on a
Sector Imager 6000 (Meso Scale Discovery) according to the manu-
facturers' instructions for the V-PLEX Human Cytokine-44 Plex Kit,
U-PLEX Human TGF-β1-Kit and R-PLEX Human Granzyme B-Kit.

### T cell stimulation
Physiological activation of human T cells was achieved with Dynabeads
Human T-Activator CD3/CD28. First, Dynabeads Human T-Activator
CD3/CD28 were resuspended in the vial. Then, the desired volume of
Dynabeads was transferred to a tube. An equal volume of buffer were
added and mixed. Next, the tube was placed on a magnet for 1 min, and
the supernatant was discarded. The Dynabeads were resuspended in a
volume of culture medium equal to the initial volume of Dynabeads
taken from the vial. Then, T cell stimulation was performed with $5 \times 10^5$
PBMC in 100−200 μl of medium in a 96-well tissue culture plate. Two
microliters of Dynabeads Human T-Activator CD3/CD28 was added to
obtain a bead-to-cell ratio of 1:1. Finally, the plate was incubated in a
humidified CO2 incubator at 37 °C for 72 h, and the cell culture
supernatant was harvested for further MSD tests.

### CyTOF examination of immune subpopulations
After PBMC collection, samples were cryopreserved. Approximately
5−6 million cells were shipped to a CyTOF service provider (Immu-
noscape, Singapore). Forty major and minor immune markers were
analyzed simultaneously (the target panel composition is detailed in
Table S2). Antibodies were labeled with an optimal combination of

metals. HD samples were run in parallel with patient samples. Samples were thawed at 37 °C, transferred into complete RPMI medium supplemented with 10% fetal calf serum (hiFCS), 1% penicillin/streptomycin/glutamine, 10 mM HEPES, and 55 μM 2-mercaptoethanol (2-ME) supplemented with 50 U/ml benzonase (Sigma) and immediately processed for staining. Since considerable variation in sample quality was observed, a sorting step was implemented for some of the samples to overcome poor sample quality, which might have resulted in higher background signals or cell loss during sample staining. Therefore, cells were stained with fluorophore-conjugated (allophycocyanin, APC) anti-human CD45 antibodies (BioLegend) and a Live/Dead (Thermo-Fisher) cell stain on ice for 20 min. Subsequently, the cells were washed twice, and live CD45-positive lymphocytes were sorted using a FAC-SAria II flow cytometry-based cell sorter (Beckton Dickinson). Sorted cells were then added to HD PBMC to reach a minimum of $3 \times 10^6$ cells per staining condition. Some methodological validation data were detailed in supplementary methods document.

For antibody staining, samples were stained with a primary fluorophore-labeled anti-TCRγδ antibody for 30 min on ice, washed twice, and then incubated with 50 μl of a metal-labeled antibody cocktail for 30 min on ice, followed by fixation in 2% paraformaldehyde in PBS overnight at 4 °C. The samples were then washed once in permeabilization buffer and barcoded with a unique combination of two distinct barcodes for 30 min on ice. The cells were washed once, incubated in cytometry buffer for 5 min, and then suspended in 250 nM iridium intercalator (DNA staining) in 2% paraformaldehyde/PBS at RT. The cells were washed, and the samples from each patient were pooled together with 1% polystyrene bead standards (EQ™ Four element calibration beads, Fluidigm) for acquisition on a HELIOS mass cytometer (Fluidigm).

The signals for each parameter were normalized based on the equilibration beads (EQ™ Four Element Calibration Beads, Fluidigm) added to each sample. Since mass cytometry provides absolute quantitation of isotopic metal labels bound to each cell, metal-conjugated antibodies that are not detected on single cells are measured as zero values. To improve the visualization of cells displayed in a compressed 2-dimensional dot plot, we randomized the signal of zero into values between 0 and 1 using R with the flow Core package; this data processing does not affect further downstream analysis. Each sample was manually debarcoded and then gated on live CD8$^+$ T cells (CD45$^+$DNA$^+$cisplatin$^-$CD3$^+$ cells) after gating out natural killer (NK) cells (CD56$^+$CD16$^+$), monocytes (CD14$^+$) and TCRγδ cells (CD3$^+$TCRγδ$^+$) using FlowJo software (TreeStar Inc.). Correlation or trend analysis was performed to correlate immune profiles with drug response. Bioinformatic analysis was performed using phenotypic profiling involving UMAP dimensionality reduction (by 'umap' in R) and clustering algorithms (PhenoGraph clustering by 'Rphenograph', an R implementation). Clusters are labeled based on scaled (x/max, limit to less than 1) median of marker normalized expression values. Some clusters are hard to define as any known cell population, so they are labeled as 'unknown'.

CyTOF QC test: if the viable cells count of the samples were lower than 1000 or if the sample is the outlier in principle component analysis (PCA) results, it means the samples failed the CyTOF QC test and were excluded from this study (PCA results were shown in Supplementary Method Document).

### Statistical analysis
All experiments were performed at least in triplicate. T-test was used to compare a single experimental group with a control group, or responders with non-responders. One-way ANOVA was used for multiple-group comparisons. SPSS software was used for clinicopathological parameter analysis, and tools such as the chi-square test were used. R software was used for immune marker level, cytokine level or immune cell frequency analyses. The two-sided significance level was set at $p < 0.05$.

### Reporting summary
Further information on research design is available in the Nature Portfolio Reporting Summary linked to this article.

## Data availability
All data generated or analyzed during this study and the dataset corresponding to the main figures is accessible in this published article (and the supplementary source, under 'Figure source'). The raw fcs files for CyTOF have been deposited in Harvard Dataverse (https://dataverse.harvard.edu/) under the Persistent Identifier: (https://doi.org/10.7910/DVN/I9TGTE). Source data are provided with this paper.

## Code availability
All code or mathematical algorithm used in this study are included in this published article (and its supplementary information files).

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

## Acknowledgements

We thank the patients and their families. The research leading to these results has received funding from the Macau Science and Technology Development Fund (Project no: 0063/2022/A2 & 001/2020/ALC), 2020 Young Qihuang scholar funded by the National Administration of Traditional Chinese Medicine, and Janssen therapeutic fund (Project code: ICD#1101175), the National Natural Science Foundation of China (82204677), Science and Technology Projects in Guangzhou (SL2022A04J00459), the 2020 Guangdong Provincial Science and Technology Innovation Strategy Special Fund (Guangdong-Hong Kong-Macau Joint Lab) (Project no: 2020B1212030006), the Start-up Research Grant of University of Macau (SRG2022-00020-FHS) and the Faculty of Health Science, University of Macau, as well as the National Natural Science Foundation of China (82204677). Science and Technology Projects in Guangzhou (SL2022A04J00459) and Technology Research Projects of State Key Laboratory of Dampness Syndrome of Chinese Medicine (no. SZ2022KF20).

## Author contributions

Contributors L.L., Y.B.C., and E.L.H.L. contributed to research design. R.Z.L. and Y.B.C. contributed to data collection. R.Z.L., X.X.F., L.Y.W., Y.W., Z.B.J., J.M.H., H.D.P., Y.F., H.M.X., F.W., H.P.R., P.W., H.S., M.F., A.N., P.G., and L.Z. analyzed and interpreted data. R.Z.L. and X.X.F. wrote the report, which was critically reviewed and revised by L.L., Y.B.C., and E.L.H.L. All authors reviewed and approved the final report.

## Competing interests

The authors declare no competing interests.
