## [Peer Review File · Nature Communications]

Longitudinal high-dimensional analysis identifies immune features associating with response to anti-PD-1 immunotherapyREVIEWER COMMENTS

Reviewer #1 (Remarks to the Author):

This paper analyzes PBMC populations and cytokines in 25 non-small cell lung cancer patients treated with anti-PD-1 immunotherapy, in an effort to find predictive biomarkers of response. The authors show differences in total CD8+ T cells that are unexpectedly higher in non-responders (NR) than in responders (R) (though there is a lot of overlap in the distributions). Further, they show that the CD101^{high}TIM3⁺ subset of CD8+ T cells have a similar difference in NR vs. R, suggesting exhausted CD8+ T cells may be the cause of the NR outcome. This is further supported by stimulated cytokine assays, which show more impressive differences between NR and R. While provocative, there are some issues with the manuscript as currently written.

Overall, the sample size is small (25 patients), and the major findings of CD8+ T cell differences between R and NR are not very impressive (highly overlapping distributions). Furthermore, the paper doesn't state the precise number of R and NR, and the number of points in the scatter plots of Figures 1, 2, and 4 seem to vary and mostly don't seem to add up to 25. The number of patients in each category needs to be clearly stated, and if any analysis uses less than the full patient set, that needs to be noted and explained.

Is the CD8+ T cell frequency difference in NR vs. R reflective of a higher absolute count of CD8's in NR's, or are there other compartments that are decreased in NR's to cause the CD8's to be proportionally higher?

There should also be a more directed look at the phenotypic differences of CD8+ T cells in NR and R. No data on basic naive/CM/EM/TEMRA distribution is given, despite appropriate markers for such analysis in the panel. There is also no data shown on the relative distribution of CD101 and TIM3 expression, just an analysis of CD8+ T cells that are positive for both. And what about LAG3 and PD-1 expression (also markers of exhaustion in their CyTOF panel, but not really mentioned here)? Indeed, the "CCT" (CD101*TIM3+ CD8's) are only about 10% of the CD8 pool (per Figure 1), so what other subsets cause the overall difference in CD8's between R and NR? All in all, the reader doesn't get a good appreciation of the true phenotypic and numerical differences in the CD8 compartment of R vs. NR.

If NR have a more exhausted CD8+ T cell compartment at baseline, could this be due to CMV status (CMV is a known driver of terminal differentiation and exhaustion of CD8+ T cells)?

Figure 1A and B: The legend needs to state if these were only baseline data.

Figures 3 and 4: Why do the NR data end at cycle 3? The Methods state that patients received 6 treatment courses; but responders have data to cycle 10, while NR stop at cycle 3.

Figure 4: Part A has the heading "Unstimulated", but the legend indicates these are plasma levels. Plasma is unstimulated, yes, but the heading is misleading, as it suggests unstimulated cultures, in contrast to Part B, which is "after T cell stimulation". Also, the text is not clear on the stimulation process. The Results refer to stimulation with "anti-CD3/CD28 or PMA" (line 139), but Figure 4 and its legend don't specify which stimulus was used. This needs to be clarified; and if both were tried, it would be good to see the comparison.

Figure S6: There are multiple subsets with the same label (e.g., "CD8+ T cells-unknown subtype") but with different distributions and p values. What does "unknown subtype" and "NA" mean, and how were these populations identified? The Methods refer to "UMAP dimensionality reduction and clustering algorithms", but there is no detail on how the clustering was performed. More information and more explanatory labeling in the Figure is needed.

Figure S7: The labels at the top of each panel are also confusing, e.g.:

What is "unknown type"?

All immune cells should be CD45+. Why are many populations listed as CD45-?

Again, details on the clustering method used to derive these populations are needed.

Table S2: Metal labels for each antibody should be listed, as well as clone and source. Also, should parameter 28 be CXCR5 (listed as XCR5)?

Line 156: LAG3 is repeated.

Reviewer #2 (Remarks to the Author):

In this work, Li et al have combined high-dimensional CyTOF and multiplex cytokine approaches to longitudinally analyze twenty-five patients diagnosed with NSCLC.

Major comments:

1. This is interesting work analyzing the changes in immune cells of NSCLC patients treated with anti PD-1 therapy using CyTOF and multiplex cytokine approaches. How was the sample size of 25 patients determined? Was there a power calculation?
2. Were there any differences based on the different histological sub types of NSCLC?
3. What stage were the lung cancer patients? Were there any differences in the PFS based on the stage of the cancer? Description of the cohort of NSCLC needs to be included in the report.
4. It is unclear how the methodology was validated. Please provide more details. Also, what are the limitations of the method? Please describe the limitations in description.
5. Details of the statistical analyses performed are missing. Please provide the details of the analyses and the results.

REVIEWER COMMENTS

Reviewer #1 (Remarks to the Author):

We deeply appreciate the positive feedback, valuable comments and constructive suggestions. We have carefully answered the questions one by one as following.

1.1. Overall, the sample size is small (25 patients), and the major findings of CD8⁺ T cell differences between R and NR are not very impressive (highly overlapping distributions).

Response:

Thank you sincerely for your comments. On one hand, through literature searching, we found that most of the reported tests for CyTOF in clinical studies have different levels of overlapping due to technical visualization variability between CyTOF and FACS¹⁻³. It was also reported by other researchers that the variation of visualization of CyTOF technology is present and the data is relatively discrete when compared with FACS detection results⁴. However, such visualization variation does not affect the accuracy and conclusion. In our study, we have also carried out careful verification to technically exclude variable reasons of artificial operation, sample transportation, sample testing. Finally, we confirmed that our results are accurate and convincing. The unavoidable overlapping in our testing data was due to patients' specificity and variability and this part has been replenished into Materials and Methods in manuscript and Supplementary Method Document.

In this study, we have performed a systematic analysis to catalog the repertoire of surface molecules expressed by immunocytes. Although high overlapping between R and NR, CD8⁺ CD101^{hi} TIM3⁺ T cells subset was statistical difference between R and NR. CD101⁺ TIM3⁺ recently have been reported to inhibit TCR signaling and impair T cell function. TIM3⁺ CD101⁺ T cells also has been proved as the terminally differentiated T cells and highly dysfunctional antigen specific CD8⁺ T cells with exhausted state. Our finding represents the first clinical study confirmed the importance of this subset at this moment, CD8⁺ CD101^{hi} TIM3⁺ T cells in Chinese NSCLC patients,

which can contribute to develop as new biomarkers and provide a critical immunotherapy target in combination of anti-PD-1 strategy in the future.

Meanwhile, during the project initiation time, our pioneer reference literature (Becher et. al. Nat Med) has included 20 patients (10 vs 10)⁵. We did level of expansion of samples from that to accommodate both research quality and feasibility. We also calculated the sample sizes through statistical power calculation and the values are over 0.7 for both cell types, CD8⁺ and CD8⁺ CD101^{hi} TIM3⁺ T cells. Therefore, the sample size of 25 patients would be sufficient considering the translational nature of such studies.

1.2. Furthermore, the paper doesn't state the precise number of R and NR, and the number of points in the scatter plots of Figures 1, 2, and 4 seem to vary and mostly don't seem to add up to 25.

Response:

Thank you for your valuable comments. The precise number was added in the manuscript page 3 line 87-88. Because 2 baseline samples failed the QC test, we have excluded 2 baseline samples (J015A and J024A, both patients are NR). Therefore, for Figures 1, 2 & 4, there are 23 spots as the bar shows the baseline statuses. These excluded samples were also noted and explained in the figure legends. Then, for the number of followed cycles samples, it mainly depended on the treatment efficacy. If the patient responses, they could further receive anti-PD-1 treatment, the samples were included in the longitudinal analysis. The detailed information including all the samples tested in this study is listed below. The total tested samples number was 102, and the total number of patients are still 25.

Responder: 11 patients

CR: J005 A-G
 PR: J013 A, C-K
 J014 A-B
 J018 A-G
 J022 A-H
 J028 A-C
 J029 A, C
 J030 A, B
 J031 A-F
 SD: J006 A-D
 J025 A-C

NR: 14 patients

PD: J002 A, B, C
 J003 A
 J004 A
 J008 A,B
 J009 A-D
 J010 A-C
 J015 B,C
 J021 A-D
 J023 A
 J024 D
 J026 A
 J033 A-C
 J034 A-C
 J035 A-C

1.3. The number of patients in each category needs to be clearly stated, and if any analysis uses less than the full patient set, that needs to be noted and explained.

Response:

We appreciated your helpful advice. As we answered above, we have excluded two baseline samples (J015A and J024A) as they failed the QC test, therefore, the number of responders vs non-responders was (11 vs 12) before the treatment.

Then, we also performed QC test for all the rest of samples from all patients, J013B, J029B and J024B & J024C were excluded due to the failure in the QC test. However, by statistical adjustment, these missing samples does not affect the final conclusion. Analysis uses less than the full patient set were noted and explained in the figure legends.

The reason for the patients not receiving the same number of immunotherapy treatment cycles is due to ethical reason. According to the guideline in NCCN, the current drug treatment should be replaced by another method if the evaluation of the current therapy is not efficacious for non-responders. Therefore, we could observe that the responders in our study received far more treatment cycles, when compared with non-responders.

2. Is the CD8⁺ T cell frequency difference in NR vs R reflective of a higher absolute

count of CD8's in NR's, or are there other compartments that are decreased in NR's to cause the CD8's to be proportionally higher?

Response:

Thanks, it is a valuable question. According to the results below, both contributing factors were presented. Firstly, we can realize that compared with the level of HD, the absolute percentage of CD8⁺ T cells increased in NR significantly. Secondly, we could observe that the frequency of CD4⁺ T cells and NK cells decreased in NR's PBMC compared with R after receiving anti-PD1 treatment. Thus, a combination of two factors contributed to the result.

3.1. There should also be a more directed look at the phenotypic differences of CD8+ T cells in NR and R. No data on basic naive/CM/EM/TEMRA distribution is given, despite appropriate markers for such analysis in the panel.

Response:

We completely agreed with your comments. In fact, we have tested various phenotypes of T cells (Figure S6), including CD8+ T cells and its differentiated phenotype CD8+ T cell memory CD45RO+, but there was no statistical significance between responders and non-responders.

Frequency in samples (Percentage of CD45RO+ in CD8+ T cells)

As for our finding phenotype of CD8+ T cells, CD8+ CD101^{high} TIM3+ T cells, it has been reported that the subset (TIM3+ T cells) was the terminally differentiated CD8+ T cell population in chronic infection models. With our unique Chinese NSCLC clinical samples, we for the first time demonstrated the prognostic value of CCT subset in

immunotherapy and we proved that TIM3⁺ alone was not enough to predict treatment efficacy and that a combinational performance of TIM3⁺ and CD101^{hi} would be needed.

Normalized expression of TIM3 (CD101) in CD8⁺ T cells

In this study, due to the limitation of detection technology, we could only detect a maximum of 40 antibodies at the same time through CyTOF, which is the maximum number of antigens and protein expression in living cells that can be detected at present. At the same time, referring to the high-quality articles published so far, we found that CD101⁺ TIM3⁺ T cells existed at the final stage of T cells differentiation in chronic infection mice models ⁶. Based on the above two reasons, we did not subdivide CD45RO cells into naive/CM/EM/TEMRA distribution ⁷ for detection this time.

3.2. There is also no data shown on the relative distribution of CD101 and TIM3 expression, just an analysis of CD8⁺ T cells that are positive for both.

Response:

Thank you sincerely for your comments. We have tested all the distributions of CD101 and TIM3 expression in different major immune cell populations as well as in subsets. However, there was no statistical significance in the results as shown below. To better illustrate our findings, we have added this result as well as explanation into manuscript, please refer main text page 4 line 95-98 and Figure S8.

Normalized expression of CD101 in major immune cell populations between R and NR

Normalized expression of TIM3 in major immune cell populations between R and NR

3.3. And what about LAG3 and PD-1 expression (also markers of exhaustion in their CyTOF panel, but not really mentioned here)?

Response:

We appreciate very much for your question. We have also tested other exhausted markers as shown below, but there was no significant difference between responders and non-responders both in major populations and subsets.

Normalized expression of PD-1 in major populations

Normalized expression PD-1 in subsets

For LAG3, after rechecking, due to limitation of 40 markers capacity of CyTOF, we have not included LAG3 in our panel, the previous table might contain typo mistakes. We have corrected the Materials and Methods part and the content of Table S2 according to the final confirmed antibody content list.

3.4. Indeed, the “CCT” (CD101^{hi}TIM3⁺ CD8’s) are only about 10% of the CD8 pool (per Figure 1), so what other subsets cause the overall difference in CD8’s between R and NR? All in all, the reader doesn’t get a good appreciation of the true phenotypic and numerical differences in the CD8 compartment of R vs. NR.

Response:

We are grateful for your suggestions. The 10% was the frequency of cluster CCT/CD45⁺. If we chose the frequency of cluster CCT/parent cell population-CD8⁺ T cells, our finding subtype “CCT” would be 35%. In order to fully demonstrate the accuracy of our data, we compared the reported literatures. First, we found that the proportion of main immune cells detected by us, such as CD4⁺ T cells and CD8⁺ T cells, was basically consistent with the proportion of reported cells, about 40% of CD4⁺ T cells and 25% of CD8⁺ T cells⁵. Then, we found that the expression of TIM3 in CD8⁺ T cells was less than 10% investigated by other researchers, which suggested that our results were in line with most of the reports and thus our data is convincing². After analyzing other CD8 subpopulation frequency in CD45, although CCT was the only subset exhibited significant difference between R and NR as shown in Figure S6, this CCT subset might be playing an essential role as predictive biomarker.

4. If NR have a more exhausted CD8⁺ T cell compartment at baseline, could this be due to CMV status (CMV is a known driver of terminal differentiation and exhaustion of CD8⁺ T cells)?

Response:

Thank you for your comments. To answer this question, we have searched many published articles and consulted with several clinical doctors. It has been reported that CMV could indeed affect the function and differentiation of T cells, especially in patients with chronic diseases and autoimmune deficiency. However, there are few cases reported in lung cancer patients. Then, based on the current diagnosis and treatment experience on Macao patients, CMV is not a routine examination indicator.

Figure 1a and b: The legend needs to state if these were only baseline data.

Response:

We are grateful for your suggestions. We have revised the figure legend in Figure 1.

Figures 3 and 4: Why do the NR data end at cycle 3? The Methods state that patients received 6 treatment courses; but responders have data to cycle 10, while NR stop at cycle 3.

Response:

Thanks so much for your questions. In clinical practice, we do hope that all cancer patients could benefit from immunotherapy treatment and survive for a long time. Unfortunately, the results of treatment in the actual process are not satisfactory. Most immunotherapy patients have a low response rate, so the duration of treatment is adjusted according to their treatment efficacy under the NCCN guidelines and clinical ethics. If the disease progresses, the immunotherapy strategy need to be adjusted. Thus, most of the non-responders received fewer treatments. Thanks for your question on the method typo part, we have revised it.

Figure 4: Part A has the heading “Unstimulated”, but the legend indicates these are plasma levels. Plasma is unstimulated, yes, but the heading is misleading, as it suggests unstimulated cultures, in contrast to Part B, which is “after T cell stimulation”. Also, the text is not clear on the stimulation process. The Results refer to stimulation with “anti-CD3/CD28 or PMA” (line 139), but Figure 4 and its legend don’t specify which stimulus was used. This needs to be clarified; and if both were tried, it would be good to see the comparison.

Response:

We are grateful for your reminding. We apologized for the misleading caused by typo of the heading, and we revised the heading of Figure 4a to specifically state that anti-CD3/CD28 stimulation is used. Actually, we have tested both approaches of stimulation, but the results after PMA stimulation unfortunately could not pass the QC due to insufficient number of cells for additional stimulation experiments (For ethical reason, only 7 ml of blood could be collected from patient for each cycle), and have to be abandoned, so only anti-CD3/CD28 stimulation is used We have revised the

Methods content in the manuscript according to the actual situation on page 10.

Figure S6: There are multiple subsets with the same label (e.g., “CD8+ T cells-unknown subtype”) but with different distributions and p values. What does “unknown subtype” and “NA” mean, and how were these populations identified? The Methods refer to “UMAP dimensionality reduction and clustering algorithms”, but there is no detail on how the clustering was performed. More information and more explanatory labeling in the Figure is needed.

Response:

We agreed with your comments. The reason that we labeled some clusters as “unknown” the same as “NA” was that we could not define the subtypes by the 40 markers we applied. However, fortunately the results showed no statistical difference between R and NR. Details on algorithm have been added to manuscript. For the labeling, we have added unique ID 1,2,3,4...for each unknown subset to differentiate them and have unified “NA” as “unknown subtype”.

For clustering method, we have added back some explanation, please refer main text page 12 line 325-329. More information and explanatory labeling were also added into the figure legend S6.

Figure S7: The labels at the top of each panel are also confusing, e.g.:

What is “unknown type”?

All immune cells should be CD45⁺. Why are many populations listed as CD45⁻?

Again, details on the clustering method used to derive these populations are needed.

Response:

We sincerely appreciate your comments. In Figure S7, we compared the subtypes between healthy donors (HD) and patients at baseline. The unknown types were same as we replied above, and we have unified the labeling methods. As for the “CD45⁻”, it is not mean “CD45 neg”, but “CD45 positive combined with other features”. Sorry for the misleading. We have revised them all in the supplementary figures.

Table S2: Metal labels for each antibody should be listed, as well as clone and source. Also, should parameter 28 be CXCR5 (listed as XCR5)?

Response:

Thank you very much for your suggestion. We have listed the metal labels below and corrected the typo in the Table S2.

Metal	Antibody	Clone	Provider
Y-89	CD45	HI30	Fluidigm
Cd-112/114	CD14	TUK4	Invitrogen
In-115	CD15	HI98	Biolegend
Ce-140	Anti-PE	PE001	Biolegend
	TCRgt-PE	5A6.E9	Invitrogen
Pr-141	CD56	NCAM16.2	BD
Nd-142	CD19	HIB-19	Biolegend
Nd-143	CD27	LG.7F9	Biolegend
Nd-144	CD3	UCHT1	bl300414
Nd-145	CD8	SK1	Biolegend
Nd-146	HLA-DR	L 243	Biolegend
Sm-147	CD4	SK3	Biolegend
Nd-148	CD45RO	UCHL 1	Biolegend
Sm-149	ICOS	C398.4A	Biolegend
Nd-150	Granzyme B	CLB-GB11	Abcam
Eu-151	CD69	FN50	Biolegend
Sm-152	CD101	BB27	eBioscience
Eu-153	KLRG1	13F2F12	eBioscience
Sm-154	CXCR5	RF8B2	BD
Gd-155	CD33	WM53	Biolegend
Gd-156	Tbet	4B10	Biolegend
Gd-157	CXCR3	MAB160	R&D
Gd-158	SAV	NA	in-housse
	FOXP3-biot	PCH101	eBioscience
Tb-159	PDL 1	29E.2A3	Biolegend
Gd-160	PD-1	eBioJ105	eBioscience
Dy-161	TIM-3	MAB2365	R&D
Dy-162	CD95	DX2	Biolegend
Dy-163	CD127	AO19D5	Biolegend
Dy-164	IgG4	Fc-UNLB	Southern Biotech
Ho-165	Ki67	B56	BD
Er-166	Ox40	MAB3388	R&D
Er-167	GATA3	TWAJ	eBioscience
Er-168	CCR7	MAB197	R&D
Tm-169	CD25	M-A251	Biolegend
Er-170	CTLA4	14D3	eBioscience
Yb-171	CD28	CD28.2	Biolegend
Yb-172	CD38	HIT2	Biolegend
Yb-173	CD39	A1	Biolegend
Yb-174	Eomes	WD1928	eBioscience
Lu-175	CD11c	B-Ly6	BD
Yb-176	CD11b	ICRF44	Biolegend
Bi-209	CD16	3g8	Fluidigm

Line 156: LAG3 is repeated.

Response:

Thanks for your careful check. We have revised it.

References:

- 1 Gubin, M. M. *et al.* High-Dimensional Analysis Delineates Myeloid and Lymphoid Compartment Remodeling during Successful Immune-Checkpoint Cancer Therapy. *Cell* **175**, 1014-1030 e1019, doi:10.1016/j.cell.2018.09.030 (2018).
- 2 Fehlings, M. *et al.* Late-differentiated effector neoantigen-specific CD8+ T cells are enriched in peripheral blood of non-small cell lung carcinoma patients responding to atezolizumab treatment. *J Immunother Cancer* **7**, 249, doi:10.1186/s40425-019-0695-9 (2019).
- 3 Huang, A. C. *et al.* T-cell invigoration to tumour burden ratio associated with anti-PD-1 response. *Nature* **545**, 60-65, doi:10.1038/nature22079 (2017).
- 4 Trussart, M. *et al.* Removing unwanted variation with CytofRUV to integrate multiple CyTOF datasets. *Elife* **9**, doi:10.7554/eLife.59630 (2020).
- 5 Krieg, C. *et al.* High-dimensional single-cell analysis predicts response to anti-PD-1 immunotherapy. *Nat Med* **24**, 144-153, doi:10.1038/nm.4466 (2018).
- 6 Hudson, W. H. *et al.* Proliferating Transitory T Cells with an Effector-like Transcriptional Signature Emerge from PD-1(+) Stem-like CD8(+) T Cells during Chronic Infection. *Immunity* **51**, 1043-1058 e1044, doi:10.1016/j.immuni.2019.11.002 (2019).
- 7 Larbi, A. & Fulop, T. From "truly naive" to "exhausted senescent" T cells: when markers predict functionality. *Cytometry A* **85**, 25-35, doi:10.1002/cyto.a.22351 (2014).

Reviewer #2 (Remarks to the Author):

Thank you respected referee for your constructive suggestions. To better answer your questions, we will explain them one by one.

Major comments:

1. This is interesting work analyzing the changes in immune cells of NSCLC patients treated with anti PD-1 therapy using CyToF and multiplex cytokine approaches. How was the sample size of 25 patients determined? Was there a power calculation?

Response:

Thank you sincerely for your comments. We have also carefully considered this question before submitting our manuscript.

During the project initiation time, our key reference literature (Becher et. al. Nat Med) has analyzed 20 patients (10 vs 10). Therefore, we did similar level of expansion from that to accommodate both research quality and feasibility. Our study also focuses on a board spectrum longitudinal analysis including both high-dimensional CyTOF and multiplex cytokine assays which has provided novel information. We also calculated the sample sizes through statistical power calculation and the values are over 0.7 for both cell types, CD8⁺ and CD8⁺ CD101^{hi} TIM3⁺ T cells. Therefore, the sample size would be sufficient considering the translational nature of such studies.

In addition, at the time we initiated the project in 2015, anti-PD1 drug was not launched in the market in China yet not until June 2018, Macau is the first special administrative region in China where has approved the use of Keytruda, thus the number of NSCLC patients who have received anti-PD1 drug is limited, however, it also represents the group of patients with the longest follow up time in the whole country. In such circumstance, we would like to report this novel and important data at a timely manner so as to benefit patients. We have performed a systematic approach to catalog the repertoire of surface molecules expressed by immunocytes and CD8⁺ CD101^{high} TIM3⁺ T cells subset was the only one of CD8⁺ T cells showing statistical difference between R and NR. It has been reported recently CD101⁺ TIM3⁺ could

inhibit TCR signaling and impair T cell function. TIM3⁺ CD101⁺ T cells has been proved as the terminally differentiated T cells and highly dysfunctional antigen specific CD8⁺ T cells with exhausted state. Our finding is the first clinical study confirmed the importance of this subset, CD8⁺ CD101^{hi} TIM3⁺ T cells, which can contribute to the field of biomarkers and immunotherapy target in combination of anti-PD-1 strategy, and hopefully could be reported in a timely manner.

2. Were there any differences based on the different histological sub types of NSCLC?

Response:

Thank you sincerely for your comments. Among 25 patients, 21 of them have adenocarcinoma histology, and 4 of them have squamous histology. As shown in Table S1, the results showed a higher proportion of responders in patients with lung squamous carcinoma (75%) compared with the proportion of responders with adenocarcinoma (38.1%). This result was interesting and was consistent with previously reported research. We have a rough analysis of the patients by our finding markers as shown below. Compared with squamous carcinoma patients, adenocarcinoma patients harbored a higher frequency of exhausted T cells at the baseline, especially in CD8⁺ CD101^{hi} TIM3⁺ T cells with p-value <0.05. These results have been added into manuscript, please refer main text page 4 line 99-101 and Figure S9.

3. What stage were the lung cancer patients? Were there any differences in the PFS

based on the stage of the cancer? Description of the cohort of NSCLC needs to be included in the report.

Response:

We really appreciate your comments. According to the patient demographics and baseline characteristics, 24 of patients have stage IV, and 1 has stage IIIB stage. The one patient with stage IIIB received same systemic treatment as stage IV patients. According to their PFS, there's no statistical differences among them.

Agreed with your suggestion, the description of the cohort of NSCLC has been added into Methods content in manuscript on page 8 line 209-220. "Total 25 patients were collected. The clinical characteristics of the patients were shown in Table S1. Among these 25 patients, there were 11 responders and 14 non-responders. The mean \pm SD age of responders was 68.4 ± 11.4 years compared with non-responders' 59.6 ± 11.8 years. Based on the different histological sub-types of NSCLC, there was a higher proportion of responders in patients with lung squamous carcinoma (75%) compared with the proportion of responders with adenocarcinoma (38.1%). According to the patient demographics and baseline characteristics, 24 of patients have stage IV, and 1 has stage IIIB stage. The one patient with stage IIIB received same systemic treatment as stage IV patients. Based on the patients' PD-L1 IHC level, 4 levels were classified. Then the duration of response time was also summarized, which showed much longer in responders (9.4 ± 3.5 months) compared with non-responders (3.6 ± 3.6 months)."

4. It is unclear how the methodology was validated. Please provide more details. Also, what are the limitations of the method? Please describe the limitations in description.

Response:

We completely agreed with you. In order to carry out comprehensive and accurate investigation by CyTOF, we carried out methodological verification from different angles.

(1) To validate the CyTOF method by FACS tests. The designed FACS panels was shown below.

Fluorochrome (MUST Aria III)	Panel1 (T cell proliferation)	Panel2 (T cell proliferation, CD38 and PD-1)	Panel3 (MDSC and CD38)	Panel4 (Treg and CD38)	Panel5 (Treg, CD38 and CD101)
	Require intra-cellular staining	Require intra-cellular staining	Only surface staining	Only surface staining	Only surface staining
FITC	Ki-67	Ki-67	CD45	CD45	CD45
PE	CD4	CD38	CD38	CD38	CD38
PerCP	CD3	CD3	CD15	CD19	CD3
PE-Cy5					
PE-CF594	CD8a	CD8a	HLA-DR		CD4
PE-Cy7	CD45	CD45	CD14		CD25
APC	CD45RA	PD-1	CD33	CD24	CD101
APC-Cy7	Fix/Viable NR2	Fix/Viable NR2	Fix/Viable NR2	Fix/Viable NR2	Fix/Viable NR2

Then, we tested three PBMC samples from healthy donors by both FACS and CyTOF. The results showed there was no statistical significance between these two methods, suggesting the methodology of CyTOF was validated.

(2) Since we needed to send PBMC samples to ImmunoScape in Singapore for CyTOF assay, we were concerned about the proportion of viable cells recovered after cryopreservation and overseas shipment. Prior to the formal detection, we also performed a test on samples from healthy people. As shown below, the recovery rates were all over 80% meeting the demanding number of cells required for CyTOF detection

HD	Frozen PBMC	Viable PBMC after thawing	Dead cells	Recovery rate
HD-1	7.50E+06	7.30E+06	< 1E+06	97.33%
HD-2	7.50E+06	6.10E+06	< 1E+06	81.33%
HD-3	7.50E+06	6.80E+06	< 1E+06	90.67%

(3) As our research was for the first time to longitudinally analyze NSCLC patients receiving anti-PD-1 treatment, we collected the samples for a long time and divided them into multiple batches for transportation. In order to exclude the influence of time on the test results, we conducted a comparative experiment on the PBMC of healthy people after a long time of storage. The results showed that there was no significant change in the results after a long time of storage as shown below.

Subsequently, after the patients were formally included and PBMCs were collected, we also conducted a monitoring test on different batches, and the results showed that the frequency of different immune populations from different batches were basically consistent.

Thus, our results are convincing, and this part has been replenished into Materials and Methods in manuscript and Supplementary Method Document.

5. Details of the statistical analyses performed are missing. Please provide the details of the analyses and the results.

Response:

Thank you so much for your careful review. According to your suggestion, we have revised the Methods part in the manuscript.

** See Nature Research's author and referees' website at www.nature.com/authors for information about policies, services and author benefits.

This email has been sent through the Springer Nature Tracking System NY-610A-NPG&MTS

REVIEWER COMMENTS

Reviewer #1 (Remarks to the Author):

The authors have undertaken some revisions and explanations to address the critiques of the first review. However, some issues remain:

1. In the legends of Figures 1 and 2, the last sentences should include the phrase "were excluded". And the nature of "the CyTOF QC test" should be explained (either here or in the Methods).
2. The question of the absolute count of CD8+ T cells was not really addressed (instead there is reference to "absolute percentage"?). But I take the point that the frequency of CD8+ T cells is increasing while the frequency of CD4+ T cells is decreasing.
3. I can infer from the responses that the "frequency in sample" labels in Figure 1 refer to frequency of all cells (or perhaps all CD45+ cells or all live cells?). This should be specifically stated in the figure, rather than stating "frequency in sample" which was not fully clear to me.
4. I'm still confused about what is shown in Figure 4a. The legend says "plasma levels" but the heading says "Before T cell stimulation". Was it plasma or cultures from unstimulated cells? If plasma as stated in the legend, please use matching wording in the Figure.

Overall, I'm still concerned that the CyTOF clustering using Phenograph yields multiple clusters labeled only as "unknown subtype". There's no attempt to say what markers these "unknown subtypes" express. The "CCT" cluster is clearly described as being CD101^{high} and TIM3⁺, but can this subset be verified by manual gating on CD8, CD101, and TIM3, and do the manually gated frequencies also show a significant difference between R and NR? The fact that CD101 and TIM3 individually do **not** show a significant difference makes me wonder.

REVIEWER COMMENTS

Reviewer #1:

We deeply appreciate the positive feedback, valuable comments and constructive suggestions. We have carefully answered the questions one by one as following and revised our manuscript accordingly.

Q1. In the legends of Figures 1 and 2, the last sentences should include the phrase “were excluded”. And the nature of “the CyTOF QC test” should be explained (either here or in the Methods).

Response:

Thank you so much for your careful check. We have modified the figure legends.

We completely agreed with you and added the following explanation in the methods part on page 12 line 329-332, detailed in Supplementary Method Document.

CyTOF QC test: If the total viable cells count of the samples were lower than 1000 or if the sample is the outlier in principle component analysis (PCA) results (as shown below), it means the samples failed the CyTOF QC test and was excluded from the following study.

Q2. The question of the absolute count of CD8+ T cells was not really addressed (instead there is reference to “absolute percentage”?). But I take the point that the frequency of CD8+ T cells is increasing while the frequency of CD4+ T cells is decreasing.

Response:

We sincerely appreciate your understanding. Meantime, we also tried to make it clearer.

We have analyzed the absolute cell counts and found the trend is similar to the frequency results that the absolute cell counts of CD8+ T cells of NR were higher than the counts of R (as shown below). However, due to the span of the study time and the reason that the samples were sent to Singapore for testing, the number of viable cells of the patients was different. Therefore, if we only look at the number of cells collected, the baseline of individual samples would be inconsistent. Thus, we chose the absolute percentage for further analysis.

Q3. I can infer from the responses that the “frequency in sample” labels in Figure 1 refer to frequency of all cells (or perhaps all CD45+ cells or all live cells?). This should be specifically stated in the figure, rather than stating “frequency in sample” which was not fully clear to me.

Response:

Thank you for your valuable comment. You are right, the percentages for each subpopulation are based on overall live CD45⁺ singlet cell counts (CD45⁺ immune cells). This sentence has been added in legends of Figure 1-3.

Q4. I’m still confused about what is shown in Figure 4a. The legend says “plasma levels” but the heading says “Before T cell stimulation”. Was it plasma or cultures from unstimulated cells? If plasma as stated in the legend, please use matching wording in the Figure.

Response:

We agreed with your comments. Sorry for our previous phasing causing misunderstanding. After double confirmation, it means the plasma levels and we revised the labeling in the Figure 4.

Q5. Overall, I’m still concerned that the CyTOF clustering using Phenograph yields multiple clusters labeled only as “unknown subtype”. There’s no attempt to say what markers these “unknown subtypes” express.

Response:

We sincerely appreciate your comments. By checking the heatmap of each marker's scaled median intensity value in each cluster (color is marker intensity median scaled value. Scaling is $(x - \min) / (\max - \min)$), we find for cluster 26 - the 'unknown' cluster, positive markers, including CD38, CTLA4, Ki67, OX40, are not cell identity markers. They can just tell the state of this cluster, for example, Ki67 for highly proliferative. Immune cell identity markers, such as CD3, CD4, CD8, CD19, CD16 and CD56, are all negative in this cluster. So we categorize this cluster as 'unknown'.

Q6. The “CCT” cluster is clearly described as being CD101high and TIM3+, but can this subset be verified by manual gating on CD8, CD101, and TIM3, and do the manually gated frequencies also show a significant difference between R and NR? The fact that CD101 and TIM3 individually do *not* show a significant difference makes me wonder.

Response:

We deeply appreciated your helpful advice. The results below, based on any cluster analysis by manual gating, showed that the absolute cell counts in NR were obviously higher than the cell counts in R.

The manual gating strategy was shown below and the cluster CCT can be verified by manual gating and there is significant difference between NR and R.

J008A (NR)

J0014A (R)

Thanks again!

REVIEWER COMMENTS

Reviewer #1 (Remarks to the Author):

Thanks to the authors for their careful attention to all my comments from the previous review. They have corrected the text and figures as requested, and have explained their approach in terms of the "absolute percentages" and "unknown subtype" clusters. The latter may still cause some confusion for readers. But I think the paper is overall much clearer now, and I'm satisfied with their revised version.

Reviewer #3 (Remarks to the Author):

In this study, a cohort of 25 patients with Non-Small Cell Lung Cancer (NSCLC) treated with immune checkpoint inhibitors (11 responders vs 14 non-responders) were analysed by CyTOF and MSD multi-cytokine with respect to phenotypes of peripheral blood immune cells over time.

The main finding is that the frequencies of CD8+ and CD8+CD101highTIM3+ Tcells were significantly correlated with poor clinical response and poor survival. In addition, longitudinal analyses showed that KLRG1 expression and cytokines were associated with response to therapy. Based on these results the authors claim that their finding may provide new biomarkers for selecting NSCLC patients to therapies.

My main reservation relates to the authors' conclusion of using their findings as a potential new biomarker. Data shown in figure 1 indicates indeed that many of the non-responders reveal higher peripheral CD8+ cells and CD8+CD101highTIM3+ Tcells. However, the range is huge and indeed many of the non-responders show lower levels of these Tcell phenotypes as compared to responders. Thus, I wonder whether this data may indeed be used as a predictive biomarker, when applied to a single patient.

How strong is the predictive effect in comparison to established biomarkers, i.e tumor and immune cell proportional score (TPS, ICS) of PD-L1 and Tumor Mutational Burden (TMB)?

Along these lines: do these peripheral Tcell populations provide independent prognostic information on OS, when PD-L1, TMB is measured along with established negative predictive factors, such as KEAP1 and STK11 mutations ?

Unless, the prognostic significance is stronger than currently established markers, the study remains correlative with only incremental progress in our understanding of immune responses to tumors and will not lead to algorithms for clinical selection of NSCLC patients for IO therapies.

REVIEWER COMMENTS:

Reviewer #3:

In this study, a cohort of 25 patients with Non-Small Cell Lung Cancer (NSCLC) treated with immune checkpoint inhibitors (11 responders vs 14 non-responders) were analysed by CyTOF and MSD multi-cytokine with respect to phenotypes of peripheral blood immune cells over time.

The main finding is that the frequencies of CD8⁺ and CD8⁺CD101^{high}TIM3⁺ T cells were significantly correlated with poor clinical response and poor survival.

In addition, longitudinal analyses showed that KLRG1 expression and cytokines were associated with response to therapy.

Based on these results the authors claim that their finding may provide new biomarkers for selecting NSCLC patients to therapies.

Response:

We deeply appreciate the positive feedback, valuable comments, and constructive suggestions. We have carefully answered the questions one by one as following and revised our manuscript accordingly.

1. The main reservation relates to the authors' conclusion of using their findings as a potential new biomarker. Data shown in figure 1 indicates indeed that many of the non-responders reveal higher peripheral CD8⁺ cells and CD8⁺CD101^{high}TIM3⁺ T cells. However, the range is huge and indeed many of the non-responders show lower levels of these T cell phenotypes as compared to responders. Thus, I wonder whether this data may indeed be used as a predictive biomarker, when applied to a single patient.

Response:

Thank you sincerely for your valuable comments. On one hand, in this study, to catalog the repertoire of surface molecules expressed by immunocytes, we have performed a systematic analysis. Although high overlapping between R and NR, CD8⁺CD101^{hi} TIM3⁺ T cells subset was statistical difference between R and NR. CD101⁺

TIM3⁺ recently have been reported to inhibit TCR signaling and impair T cell function. TIM3⁺ CD101⁺ T cells also has been proved as the terminally differentiated T cells and highly dysfunctional antigen specific CD8⁺ T cells with exhausted state¹. Our finding represents the first clinical study confirmed the importance of this subset at this moment, CD8⁺ CD101^{hi} TIM3⁺ T cells in Chinese NSCLC patients, which can contribute to develop as new biomarkers and provide a critical immunotherapy target in combination of anti-PD-1 strategy in the future.

On the other hand, through literature searching, we found that most of the reported tests for CyTOF in clinical studies have different levels of overlapping due to technical visualization variability between CyTOF and FACS²⁻⁴. Thus, FACS was applied for validation. The results, based on any cluster analysis by manual gating, showed that the absolute cell counts in NR were obviously higher than the cell counts in R. The manual gating strategy was shown below and the cluster CD8⁺ CD101^{hi} TIM3⁺ T cells can be verified by manual gating and there is significant difference between NR and R.

Moreover, our study is a longitudinal analysis on blood biopsies with over 30 months follow-up time. Therefore, compared with the detection at a single time point, our results better illustrated the stability of CD8⁺ CD101^{hi} TIM3⁺ T cells subset for prediction of immunotherapy over time.

2. How strong is the predictive effect in comparison to established biomarkers, i.e tumor and immune cell proportional score (TPS, ICS) of PD-L1 and Tumor Mutational Burden (TMB)?

Response:

We appreciate very much for your question. At the time we started this study in 2017, it was clinically recommended that the patients receiving immunotherapy measure PD-L1 expression in tumor tissues and the data was shown in Table S1. Following your suggestion, Chi-Squared Test was applied to measure the relevance and dependency between PD-L1 IHC level and the response statue of immunotherapy. The results showed that the Chi-square value was 0.749 with p-value 0.668, indicating the expression of PD-L1 in the tumor tissues could not reflect the response state.

		PDL1 IHC level				
		>49%	1-49%	<1%	Total	
Response state	Responder	Count	6	2	3	11
		Expected Count	6.0	1.4	3.7	11.0
		In response state%	54.5%	18.2%	27.3%	100.0%
	Non-responder	Count	7	1	5	13
		Expected Count	7.0	1.6	4.3	13.0
		In response state%	53.8%	7.7%	38.5%	100.0%
Total	Count	13	3	8	24	
	Expected Count	13.0	3.0	8.0	24.0	
	In response state%	54.2%	12.5%	33.3%	100.0%	

	Value	df	Asymptotic Significance (2-sided)
Pearson Chi-Square	0.749	2	0.688

We also tested the PD-1 in blood cells by CyTOF, but there was no significant difference between responders and non-responders both in major populations and subsets.

Normalized expression of PD-1 in major populations

Normalized expression PD-1 in subsets

As for TMB, not all the patients could take the test because of the cost, which is also an important issue for patients. In our patients, 8 patients of responder and 7 patients of non-responder received TMB test. Summarized in the table below, TMB was also not a predictor of response state for these patients.

		TMB			
		High	Low	Total	
Response state	Responder	Count	1	7	8
		Expected Count	1.1	6.9	8.0
		In response state%	12.5%	87.5%	100.0%
	Non-responder	Count	1	6	7
		Expected Count	0.9	6.1	7.0
		In response state%	14.3%	85.7%	100.0%
Total		Count	2	13	15
		Expected Count	2.0	13.0	15.0
		In response state%	13.3%	86.7%	100.0%
		Value	df	Asymptotic Significance (2-sided)	
Pearson Chi-Square		0.01	1	0.919	

Thus, while other predictors were not available or accurate, the cluster CD8⁺ CD101^{hi} TIM3⁺ T cells could still independently indicate the patients who would not respond to immunotherapy and were stable over time as well as treatment.

3. Along these lines: do these peripheral T cell populations provide independent prognostic information on OS, when PD-L1, TMB is measured along with established negative predictive factors, such as KEAP1 and STK11 mutations?

Response:

Thank you for your comments. To answer this question, we have searched many published articles and consulted with several clinical doctors. It has been reported that *KEAP1* and *STK11* mutations are prognostic, not predictive, biomarkers of anti-PD-1/and anti-PD-L1 therapy in patients with NSCLC ⁵. We believe that *KEAP1* and *STK11* mutations are valuable indicators for patients to benefit from immunotherapy.

However, in clinical practice, not all the patients could accept the cost of the test. Then, with the results above, the cluster CD8⁺ CD101^{hi} TIM3⁺ T cells could provide independent predictive and prognostic information of response state and clinical outcomes. By literature research and preliminary bioinformatic analysis, there exists mechanism relationship between KEAP1 and STK11 mutations with T cell differentiation, which is worth further investigation. So far, based on the current diagnosis and treatment experience on Macao patients, KEAP1 and STK11 mutations are not routine examination indicators.

4. Unless, the prognostic significance is stronger than currently established markers, the study remains correlative with only incremental progress in our understanding of immune responses to tumors and will not lead to algorithms for clinical selection of NSCLC patients for IO therapies.

Response:

We fully understand your concern and appreciate your comments. Through the above additional data and analysis, we hope we can answer your enquiries. We believe our paper not only provide a predictive and prognostic biomarker for immunotherapeutic response, but also provide a promising target for combinational treatment. By conducting pilot studies on this cluster with better prognostic value than conventional markers, particularly important, it is the first hand high dimensional and longitudinal analysis on Chinese NSCLC cohort, we found the results so far are promising and timely report to the public and clinician is important to facilitate the development on this field. We will optimize the plans of following research with your valuable comments.

In the conclusion part, we have modified the description in line 203-206 on page 8.

Thanks again!

References:

- 1 Hudson, W. H. *et al.* Proliferating Transitory T Cells with an Effector-like Transcriptional Signature Emerge from PD-1+ Stem-like CD8+ T Cells during Chronic Infection - ScienceDirect. *Immunity* (2019).
- 2 Gubin, M. M. *et al.* High-Dimensional Analysis Delineates Myeloid and Lymphoid Compartment Remodeling during Successful Immune-Checkpoint Cancer Therapy. *Cell* **175**, 1014-1030 e1019, doi:10.1016/j.cell.2018.09.030 (2018).
- 3 Fehlings, M. *et al.* Late-differentiated effector neoantigen-specific CD8+ T cells are enriched in peripheral blood of non-small cell lung carcinoma patients responding to atezolizumab treatment. *J Immunother Cancer* **7**, 249, doi:10.1186/s40425-019-0695-9 (2019).
- 4 Huang, A. C. *et al.* T-cell invigoration to tumour burden ratio associated with anti-PD-1 response. *Nature* **545**, 60-65, doi:10.1038/nature22079 (2017).
- 5 Papillon-Cavanagh, S., Doshi, P., Dobrin, R., Szustakowski, J. & Walsh, A. M. STK11 and KEAP1 mutations as prognostic biomarkers in an observational real-world lung adenocarcinoma cohort. *ESMO Open* **5**, doi:10.1136/esmoopen-2020-000706 (2020).

REVIEWERS' COMMENTS

Reviewer #3 (Remarks to the Author):

This is a revised manuscript addressing some of the reviewers' concerns, but major issues remain unclear.

I believe my first concern was addressed appropriately. The authors are right with respect to the huge methodological differences that were observed with CyTOF test, and hence clinical application in terms of useful predictive biomarker tests have not been established by this method. Still the authors make it plausible that the observed differences in CD8+/CD101high/TIM3+ are statistically significant.

My second and third concerns remain unsolved and have not been addressed. I do believe the authors are right when stating that not the patients should pay the diagnostic costs, but in research the investigators should do so. Thus, the statement that the patients were not able to pay molecular testing and therefore, the results here may simply reflect immune-resistant phenotypes of lung cancers remains unaddressed.

The statement that KEAP1 and STK11 mutations are prognostic but not predictive for response to immune therapies is clearly wrong. There are many published studies and hence KEAP1 and STK11 testing has become standard in Molecular Lung Cancer Diagnostics (i.e. see Marinelli D, Ann Oncol, 2020)

REVIEWER COMMENTS:

Reviewer #3:

This is a revised manuscript addressing some of the reviewers' concerns, but major issues remain unclear.

I believe my first concern was addressed appropriately. The authors are right with respect to the huge methodological differences that were observed with CyTOF test, and hence clinical application in terms of useful predictive biomarker tests have not been established by this method. Still the authors make it plausible that the observed differences in CD8+/CD101high/TIM3+ are statistically significant.

Response:

We deeply appreciate your understanding and positive feedback of our statically significant findings, thanks for the valuable comments, and constructive suggestions for improving our manuscript.

My second and third concerns remain unsolved and have not been addressed. I do believe the authors are right when stating that not the patients should pay the diagnostic costs, but in research the investigators should do so. Thus, the statement that the patients were not able to pay molecular testing and therefore, the results here may simply reflect immune-resistant phenotypes of lung cancers remains unaddressed.

Response:

We appreciate your understanding and acknowledge your concerns. In our study, we have spanned over 30 months and the project started in 2018. Dated back to 2018, we faced difficulties and limitations as we are the leading and pioneer group to initiate this kind of study in Macau, due to this kind of early initiative, we could only enroll lung cancer patients with PD-L1 expression levels, as assessed by IHC, as a biomarker for immunotherapy treatment. Although we understood tumor mutation burden (TMB) is another potential predictor of immunotherapy efficacy, it was not incorporated in our study, as it was not routinely required to be performed for all patients at that time

according to Macau Medical regulation. As a result, in clinical practice, we prioritize PD-L1 expression over TMB for immunotherapy drug indication. Despite not covering all possible factors, these patients do reflect immune resistance phenotype, who are in high demand for new therapy, we believe that our findings offer valuable insights into the immunotherapy treatment and resistance mechanism of lung cancer patients. Hope these can be reported in a timely manner.

The statement that KEAP1 and STK11 mutations are prognostic but not predictive for response to immune therapies is clearly wrong. There are many published studies and hence KEAP1 and STK11 testing has become standard in Molecular Lung Cancer Diagnostics (i.e.see Marinelli D, Ann Oncol, 2020)

Response:

Thank you sincerely for your valuable correction. We have added these two mutations in the Introduction section to help clarify the research progress in this area and to provide potentially valuable targets for future clinical application.